# Exploring the Role of DARPP-32 in Addiction: A Review of the Current Limitations of Addiction Treatment Pathways and the Role of DARPP-32 to Improve Them

**Megan R. Greener \*** and **Sarah J. Storr**

Biodiscovery Institute Phase 3, Entrance 2, Building 43, University of Nottingham, University Park, Nottingham NG7 2RD, UK

\* Correspondence: megan.greener@nottingham.ac.uk

**Abstract:** We are amidst a global addiction crisis, yet stigmas surrounding addiction counterintuitively prevail. Understanding and appreciating the neurobiology of addiction is essential to dissolve this stigma and for the development of new pharmacological agents to improve upon currently narrow therapeutic options. This review highlights this and evaluates dopamine-and-cAMP-regulated phosphoprotein, Mr 32 kDa (DARPP-32) as a potential target to treat various forms of substance abuse. Despite the proven involvement of DARPP-32 in addiction pathophysiology, no robust investigations into compounds that could pharmacologically modulate it have been carried out. Agents capable of altering DARPP-32 signalling in this way could prevent or reverse drug abuse and improve upon currently substandard treatment options.

**Keywords:** opioids; alcohol; nicotine; stimulants; cannabinoids; addiction; dependence; treatment; DARPP-32

## 1. Introduction

Recent drug reports show that half a million deaths worldwide are annually attributed to drug use, and even more deaths occur through indirect association to substance abuse when considering use as a risk factor for premature death [1]. Similar reports on international alcohol consumption suggest harmful use of alcohol causes around 3 million deaths per year [2]. Tobacco use results in around 8 million deaths a year; more than 7 million from tobacco use directly and approximately 1.2 million from second-hand smoke exposure [3]. These already concerning figures are likely to increase in response to the negative impact of the COVID-19 pandemic on the mental health of the general population, and resultingly higher levels of drug and alcohol consumption [4]. Developing issues such as the emergence of illicit drugs sold as benzodiazepines or synthetic cannabinoids, increasing variability in the purity of recreational drugs and the continuing burden of addiction to prescribed medication could augment this further [5]. Conclusively, we are amid an international addiction crisis.

Counterintuitively there is a prevailing stigma surrounding addiction treatment. Commonly public perceptions can fail to legitimize addiction as a disease or can place blame on the individual suffering from the disease itself [6]. For this individual, there are often additional obstacles to overcome before seeking clinical help; they must first overcome the anticipation of stigma and judgement. The result of this is often a reduction in accessing treatment and resultingly higher rates of substance abuse related hospitalisations [7]. More broadly, this stigma translates to a disproportionate lack of addiction research and financial underinvestment into addiction treatments. The result is often care from non-specialist health care professionals and sub-optimal treatment pathways. Limited options of suitable pharmacological interventions heighten the crisis further [8]. Addiction is a key public health issue, and one that requires validation as such. It is vital that the stigma of addiction

is dissolved, appreciation of the proven neurobiological mechanisms of addiction are acknowledged and efforts are put towards the discovery of new pharmaceutical agents that can improve upon currently narrow treatment options.

## 2. Current Pharmacological Treatment

Evidence-based treatment of addiction currently relies on a combination of behavioural and pharmacological interventions. Comprehensively, the treatment of addiction must be in lieu of a drug-free lifestyle and restoration of productive functioning in the individual. The goal of pharmacological intervention specifically is to achieve drug abstinence [9].

### 2.1. Pharmacological Treatment of Alcohol Addiction

Disulfiram, acamprosate and naltrexone are the most used from the narrow list of pharmacological agents licensed to treat alcohol use disorder (AUD) [10]. Other pharmacotherapies exist to assist a recovering AUD patient, such as the use of benzodiazepines in alcohol withdrawal or thiamine in alcoholism-associated deficiencies [11]. The three listed are the only pharmacological options to treat AUD itself, rather than treating associated symptoms of an individual attempting to achieve abstinence alone.

The longest-standing drug of this list is disulfiram, having been used in the treatment of AUD since 1951 despite only moderate clinical effectiveness [12]. The basis of disulfiram use resembles aversion therapy; it induces an acute physical reaction of nausea and vomiting in response to alcohol intake [13]. Resultingly, disulfiram use can be limited by a lack of adherence, a negative public perception and effectiveness is dependent on subjects who are committed to abstinence.

There are discrepancies in the application of acamprosate to treat AUD. Firstly, the mechanism in which acamprosate can reduce alcohol intake is not fully understood [14]. Secondly, there are contrasting conclusions as to whether acamprosate is efficacious [15–17]. Whilst some studies claim acamprosate effectively reduces alcohol intake, others report no evidence of increased efficacy. What is clear throughout investigations is that acamprosate as a single agent (without combination of other pharmacotherapy or behavioural intervention) has a substandard effect on reduction in alcohol use.

Naltrexone is an opioid antagonist with extensive evidence that supports use in reduction in heavy drinking. However, the efficacy of naltrexone is less evident in terms of achieving complete abstinence and hepatotoxicity issues from oral preparations of naltrexone that require liver function monitoring present another obstacle [18].

### 2.2. Pharmacological Treatment of Opioid Addiction

Opioid use disorder (OUD) refers to the abuse and dependence of the illicit opioid heroin and those available legally. Commonly abused examples of the latter include fentanyl, oxycodone, morphine and codeine. An alarming rise in opioid misuse has occurred over recent years culminating in a greater number of people using opioids and a sharp increase in death rates predominantly from overdose [19].

Commonplace treatment for OUD is through opioid maintenance therapy (OMT). One of two potent and long-acting opioid agonists methadone or buprenorphine (often combined with naloxone, an opioid antagonist added for overdose prevention) are given to patients through a controlled, daily dosing regimen. The principle behind this is prolonged stimulation of opioid receptors that relieves withdrawal symptoms and avoids cravings without the experience of euphoria [20].

Studies investigating methadone and buprenorphine as maintenance therapy agents concluded that they reduced opioid consumption and increased treatment retention when compared with non-pharmacological therapy or placebos, respectively [21,22]. Despite the success of these parameters, there are drawbacks and limitations to OMT. One difficulty faced is the detoxification from methadone or buprenorphine themselves; high rates of opioid relapse can occur during this period and effective detoxification regimens are yet to be perfected [23]. Polydrug abuse amongst OMT patients is another issue, with reports

of misuse of OMT medication and confirmed reduction in retention rates in response [24]. Due to the potential of abuse of the medications used in OMT, and reported increased mortality from their use, there are stringent regulations to delivery of the service [25]. From the constant government scrutiny of OMT in the United States to the requirement of supervision for consumption in the United Kingdom, governing the complex policies involved with OMT creates an additional barrier to the delivery of care [26,27].

### 2.3. Pharmacological Treatment of Nicotine Addiction

Pharmacotherapy interventions for smoking cessation centre around reduction in addiction to nicotine; the primary psychoactive substance and psychostimulant in tobacco [28]. A psychostimulant refers to a psychotropic substance that can alter the central nervous system [29]. To combat this, nicotine replacement therapy (NRT) is most widely used [30]. The mechanism of action of NRT is via stimulation of nicotinic receptors in the ventral tegmental area (VTA) of the brain; dopamine release and peripheral actions of nicotine result in lower nicotine withdrawal symptoms in those attempting to quit [31]. NRT has been consistently proven to increase the likelihood of a successful quit attempt; recent reports suggest by 50–60% [32].

Other commonly used pharmacotherapies in tobacco addiction are varenicline and bupropion. Varenicline, a partial nicotine agonist, and bupropion, originally an anti-depressant, both relieve cravings and withdrawal symptoms. Meta-analysis of 14 varenicline and 36 bupropion trials showed, respectively, that abstinence rate can double and quit rate significantly increases through use of these pharmacological agents [33].

### 2.4. Pharmacological Treatment of Cannabinoid Addiction

The most notable and prevalent components of the cannabis plant are $\Delta^9$-tetrahydro-cannabinol (THC), a psychotropic cannabinoid, and cannabidiol (CBD), a non-psychoactive cannabinoid [34]. Owing to the psychoactive tendencies of THC, cannabis is the third most common substance of abuse in use (after alcohol and nicotine). Of these multi-million users, 10% are at risk of life-time dependence [35]. In contrast, the weighting of cannabis use disorder (CUD) as a significant health condition is underacknowledged, especially in comparison to other abusable substances. Cannabis use has a multitude of acute mental and physical adverse effects including impairment of coordination, hyperemesis and anxiety with suicidal tendencies [36]. Chronically, exacerbation or attainment of mental illness and psychotic disorders has been extensively documented [37]. Owing to this, CUD is equally as problematic to healthcare as other forms of substance dependence.

Preclinically, substitution therapy was trialled with oral cannabinoid agonist dronabinol which assisted in the relief of withdrawal symptoms but failed to prevent relapse [38]. Use of the cannabinoid receptor antagonist rimonabant was an alternative approach, however intolerable psychiatric side effects prevented this agent from progressing to applicable treatment [39]. Increasing political agendas to legalise the use of cannabis for both medical and recreation reasons are likely to increase global usage and resultingly harm [40]. Whilst attempts to implement CBD in routine medical practice are increasing everywhere from chronic pain to the treatment of CUD itself (where more research is needed to confirm safety and efficacy), there are still currently no licensed pharmacological agents for CUD [41].

### 2.5. Pharmacological Treatment of Psychostimulant Addiction

The somewhat effective therapeutic options for opioid and alcohol abuse (and nicotine, a psychostimulant itself) are not mirrored in the treatment of psychostimulant abuse broadly. Whilst nicotine and caffeine are more prevalent, psychostimulant abuse generally refers to illicit psychostimulants including cocaine and amphetamines, the latter a group of molecules including D-amphetamine (AMPH), methamphetamine (METH), and 3,4-methylenedioxymethamphetamine (MDMA) [42]. Cocaine use has been a rising concern for over a decade, particularly in the United States and Europe, where estimated users are 2 million and 3.5 million, respectively [43,44]. Globally, illicit psychostimulants are

widely used with an estimated 18.2 million cocaine users and 34.2 million amphetamine users [45]. Despite this, there are currently no therapeutic options licensed to treat cocaine or amphetamine addiction specifically and treatment is usually reliant on behavioural interventions alone [46].

A similar approach to nicotine and opioid dependence treatment is used to attempt to combat this issue. This involves off-label use of prescription psychostimulants such as amphetamine salts or methylphenidate to prevent withdrawal [47]. However, abuse of prescription amphetamines is increasing in commonality. In the United States alone, prescription rates in young adults and adolescents tripled between 2005 and 2014 giving rise to issues such as patient misuse and prescription exchange [48]. Due to a lack of addictive properties, the psychostimulant modafinil has been extensively studied as a potential treatment option. Contrasting results in efficacy (especially when considering poly-drug dependencies) have limited use [49]. Hence, the risk of misuse and diversion is a huge limitation for this therapy.

### 2.6. Attempts to Improve Pharmacological Treatment Options

Efforts to improve upon treatment of alcohol and opioid dependence has recently centred around intramuscular depot formulations of naltrexone proven to reduce hepatotoxicity issues and improve adherence [50]. However, depot naltrexone has shown no superiority in improving abstinence in recent studies and the extent to which adherence is increased is questionable [51]. Increasing understanding of the endocannabinoid system in cannabis dependence has provided a rationale for influencing this pharmacologically. Fatty acid amide hydrolase (FAAH) is an endocannabinoid synthetizing enzyme currently showing promise as a pharmacological target for CUD. FAAH-inhibitors can influence this endocannabinoid system and novel agent PF-04457845 has been proven to reduce cannabis withdrawal and reduce relapse without adverse side effects [52]. This promising lead could be a potential pharmacological agent for CUD, if it can be separated from preconceptions of other previously tested, unsafe FAAH-inhibitors [53].

Improvement attempts for psychostimulant addiction (as well as heroin) have stemmed from the coupling of narcotics to immunogenic proteins to stimulate immune response; 'addiction vaccines'. However, no such pharmacological option has been licensed for clinical use despite decades of research efforts [54].

The anomaly here is in the treatment of nicotine dependence. Despite efficacious, evidence-based options for pharmacological treatment, tobacco consumption is still an evident public health issue. This highlights adherence as a huge issue in addiction treatment, alongside the need for improvement of combined pharmacological and behavioural efforts.

The limitations of the currently approved pharmacological agents are clear. Most notably, the lack of any available pharmacological treatment options for psychostimulant or cannabinoid abuse represents an enormous failure in the healthcare system and a definite gap in knowledge.

## 3. The Neurobiology of Addiction

Reduction in stigma and development of new therapeutic agents will rely on an acknowledgement of the neurobiological mechanisms that occur throughout the addiction process; how does a person transition from use of a substance to complete dependence on it? Whilst all drugs of abuse will inflict distinct neurobiological responses, the mechanism of addiction to each is analogous.

Our current understanding of these drug-induced pathologies is based on decades of research which were predominantly founded upon animal models of addiction where investigations into both molecular biology and resulting phenotypic behaviour were utilized [55]. An over-simplified clarification of this process is the grouping of addiction into three progressive stages; intoxication, withdrawal and anticipation (craving). Distinct neurobiological mechanisms have been documented that parallel each of these stages through both animal and human imaging studies [56].

During intoxication, all drugs of abuse increase dopamine concentration within the nucleus accumbens and dorsal striatum in the brain. This is the mesolimbic system, a key integrator of reward neurocircuitry. Hence, increased dopamine levels produce the rewarding effect associated with consumption of drugs or alcohol [57].

Normal physiological levels of dopamine within the prefrontal cortex activate $D_2$ receptors and, to a lesser extent owing to a lower binding affinity, $D_1$ receptors. High dopamine concentrations in response to substance abuse results in greater activation of the $D_1$ receptor. This is vital for both reward (through pathways modulating the striatum and cortex) and for conditioning processes that are significant to later stages of addiction (through mechanisms involving the amygdala, medial orbitofrontal cortex and hippocampus) [58].

During withdrawal, the same neuropharmacological mechanisms that stimulate reward are altered to result in a negative state that opposes prior positive effects during intoxication. Dopaminergic transmission is lower in an addicted brain going through withdrawal. As a result, the euphoria experienced during initial intoxication is diminished (even with repeated exposure to the substance). This lack of reward can translate further to result in an individual becoming less motivated by activity that was once deemed rewarding [59]. Furthermore, the resulting changes to the extended amygdala in the forebrain in response to repeated drug exposure causes activation of brain and pituitary stress systems. Increased extra-cellular corticotropin-releasing factor (CRF) in the amygdala regions of the brain gives rise to dysphoria in the individual, which physically manifests as increased stress and a prolonged negative state associated with withdrawal [60]. The resulting compulsion of this is a key element of the transition from drug use to dependence; the urge to use the substance to alleviate symptoms of this transient dysphoria, not simply for the reward.

Continual dopaminergic transmission results in alterations in neuroplasticity (the remodelling and reorganisation of neural networks in response to stimuli). The result is changes to midbrain dopaminergic neurons and their projections into the nucleus accumbens and dorsal striatum. The behavioural inflexibility and drug salience associated with this is known as conditioning. This is the route of the third stage; overwhelming cravings [61]. Neurotransmitters such as endogenous opioids, GABA and serotonin, as well as noradrenergic and cholinergic pathways are influenced by drugs of abuse. These neurotransmitters modulate the dopaminergic pathway and thus alter the mechanisms discussed [62].

Glutamate is also a key integrator of addiction neurobiology. The mesocorticolimbic dopamine system is closely related to glutaminergic structures, and glutamate has been proven to affect the dopaminergic system. Glutaminergic input increases the activity of dopaminergic cells and enhances dopamine release in the nucleus accumbens, augmenting the reward pathway. It has also been suggested that the prefronto-accumbal glutamatergic pathway contributes to the mediation of reward [63]. Furthermore, glutaminergic transmission is heavily involved in sensitisation; the associated increase in incentive salience with repeated drug-taking [64]. This is evident through changes in glutaminergic transmission that occur in response to sensitizing treatment schedules of drugs of abuse, likely through activation of $D_1$ receptors which augments glutaminergic activity through increased expression of NMDA receptors post-synaptically [65].

## 4. Dopamine- and cAMP-Regulated Phosphoprotein, Mr 32 kDa (DARPP-32)

DARPP-32 is an integrator of dopamine and glutamate [66]. Hence, it is an interesting potential target in the pursuit of improving current pharmacological treatment options for addiction.

### 4.1. DARPP-32 Discovery

The late Paul Greengard discovered DARPP-32 during his pioneering work that proved the same mechanisms used in the endocrine system are used for communication between nerve cells [67]. Within glycogenolysis, for example, epinephrine binds to a

G-protein coupled receptor (GPCR) causing a subunit to activate adenylyl cyclase and result in an increased cytosolic concentration of cAMP (the second messenger). cAMP activates protein kinase A (PKA) and this causes a cascade of phosphorylation that results in decreased glycogen synthesis and increased glycogen breakdown (see Figure 1) [68].

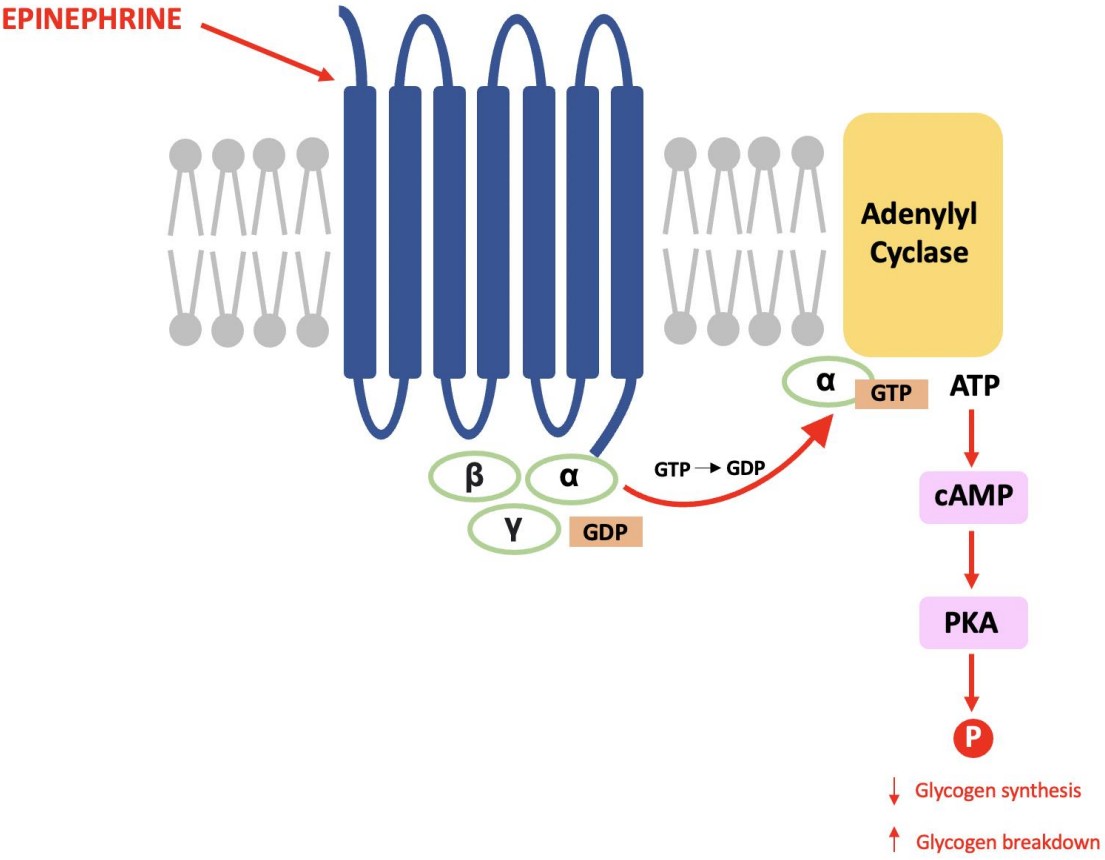

**Figure 1.** The glycogenolysis pathway: This is a second messenger pathway in which epinephrine binds to a GPCR, activates PKA and stimulates a cascade of phosphorylation that decreases glycogen synthesis and increases glycogen breakdown.

In DARPP-32 modulation, dopamine binds to $D_1$ receptors located in the striatum and causes a $G_s$ subunit to interact with adenylyl cyclase and result in the same pathway; increased intracellular cAMP, activation of PKA and, in this case, phosphorylation of DARPP-32 at the threonine-34 ($Thr^{34}$) residue [69]. The result of this pathway is the conversion of DARPP-32 into a potent inhibitor of protein phosphatase-1 (PP-1) (see Figure 2). PP-1 is a multifunctional protein affecting a variety of signalling pathways, making DARPP-32 an effector of downstream changes in physiological function and a promising target for pharmacological intervention [70].

### 4.2. DARPP-32 Phosphorylation

As discussed in 4.1, DARPP-32 is phosphorylated via PKA at the $Thr^{34}$ residue. This occurs primarily through the actions of dopamine (and $D_1$-selective agonists) on striatal neurons expressing the $D_1$ class of receptors, such as within striatonigral subpopulations [71]. Adenosine acting on $A_{2A}$-expressing regions such as within striatopallidal neurons has the same effect; increased activity of adenylyl cyclase to stimulate cAMP formation and activate cAMP-dependent PKA [72]. The effect of these two neurotransmitters is additive as they both activate the cAMP/PKA/DARPP-32 signalling pathway and cause PP-1 inhibition [73]. In contrast, $D_2$-receptor activation via dopamine and $D_2$-selective agonists reduces levels of DARPP-32 phosphorylation at the $Thr^{34}$ residue, through adenylyl cyclase inhibition that results in decreased PKA activity [74]. As PKA regulates a variety of cAMP-

dependent physiological processes, the ability to act as either a PKA or PP-1 inhibitor gives rise to the unique switch function of DARPP-32.

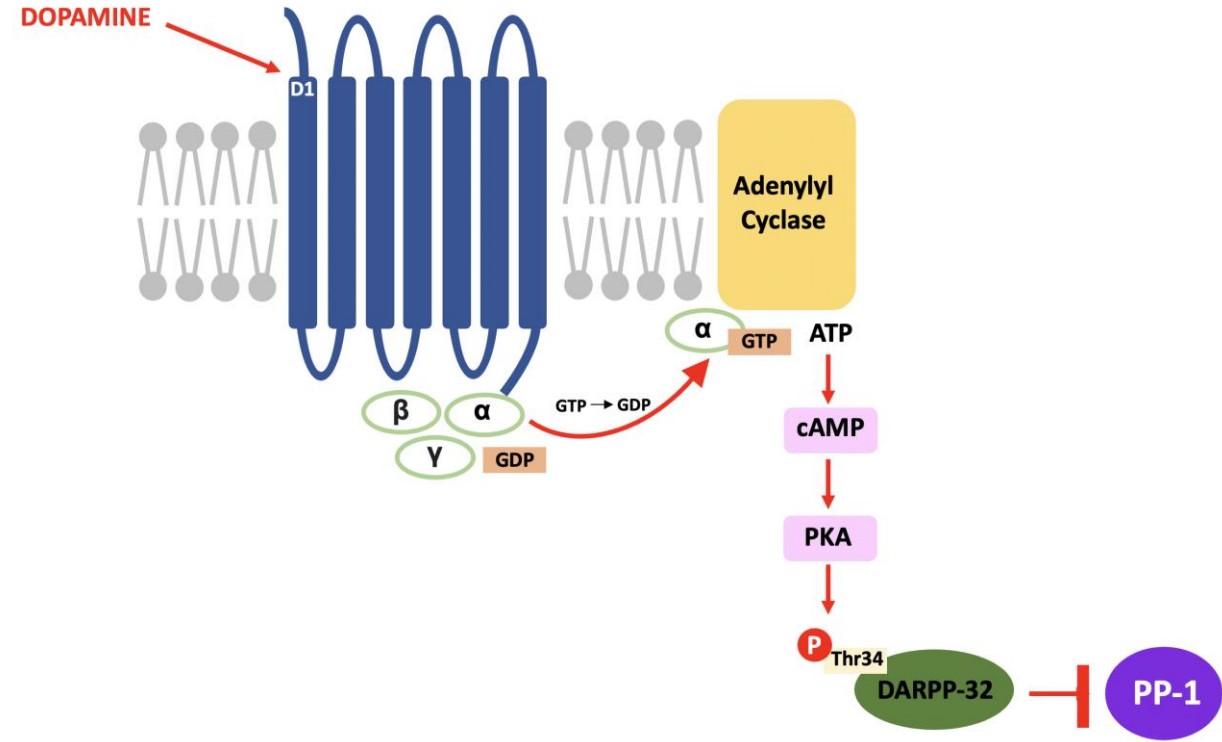

**Figure 2.** The $D_1$/DARPP-32/PKA pathway: Dopamine binds to D1 receptors, activates PKA and stimulates a cascade of phosphorylation that results in the inhibition of PP-1.

Cyclin-dependent kinase 5 (CDK-5) phosphorylates DARPP-32 at the threonine-75 (Thr$^{75}$) residue [75]. This prevents the actions of PKA at the Thr$^{34}$ site [76]. Opposingly, casein kinase I (CK1) and casein kinase II (CK2) act to attenuate the actions of DARPP-32 as a PP-1 inhibitor [77]. CK2 phosphorylates DARPP-32 at Serine-97 (Ser$^{97}$) and increases the efficiency of Thr$^{34}$ phosphorylation whilst phosphorylation of Serine-130 (Ser$^{130}$) via CK1 acts to inhibit protein phosphatase-2B (calcineurin) [78,79]. Calcineurin and protein phosphatase-2A (PP-2A) act synergistically to dephosphorylate DARPP-32 at Thr$^{34}$ [80]. Therefore, reducing calcineurin-dependent dephosphorylation of DARPP-32 would increase levels of Thr$^{34}$-phosphorylated DARPP-32.

Glutamate can also regulate DARPP-32 phosphorylation. Acting at both NMDA and AMPA receptors, glutamate causes calcium-dependent activation of calcineurin to result in dephosphorylation of DARPP-32. Opposingly, glutamate at I mGlu-5 receptors (mGlu5) potentiates cAMP formation coupled to $A_{2A}$ receptors and increases Thr$^{34}$ phosphorylation. It can also increase phosphorylation at Ser$^{130}$ and Thr$^{75}$ through group I mGlu receptors [81]. The ability of DARPP-32 to alter downstream signalling depending on phosphorylation site is indicative of the central role it plays in signal transduction (see Figure 3).

*4.3. DARPP-32 Localisation*

Through immunohistochemistry investigations, expression of DARPP-32 has been identified in the brain, adrenal medulla, kidney, and parathyroid cells [82]. However, immunocytochemistry localization experiments and biochemical studies proved that DARPP-32 is predominantly localised in medium spiny neurons (MSNs) within the striatum [83]. The striatum is a site of major dopaminergic innervation within the central nervous system. The dorsal striatum (caudate nucleus and putamen) receives dopaminergic input from the substantia nigra pars compacta that contributes to coordination and response, whilst the

ventral striatum (nucleus accumbens) is innervated from the VTA and contributes to the reward pathway [84].

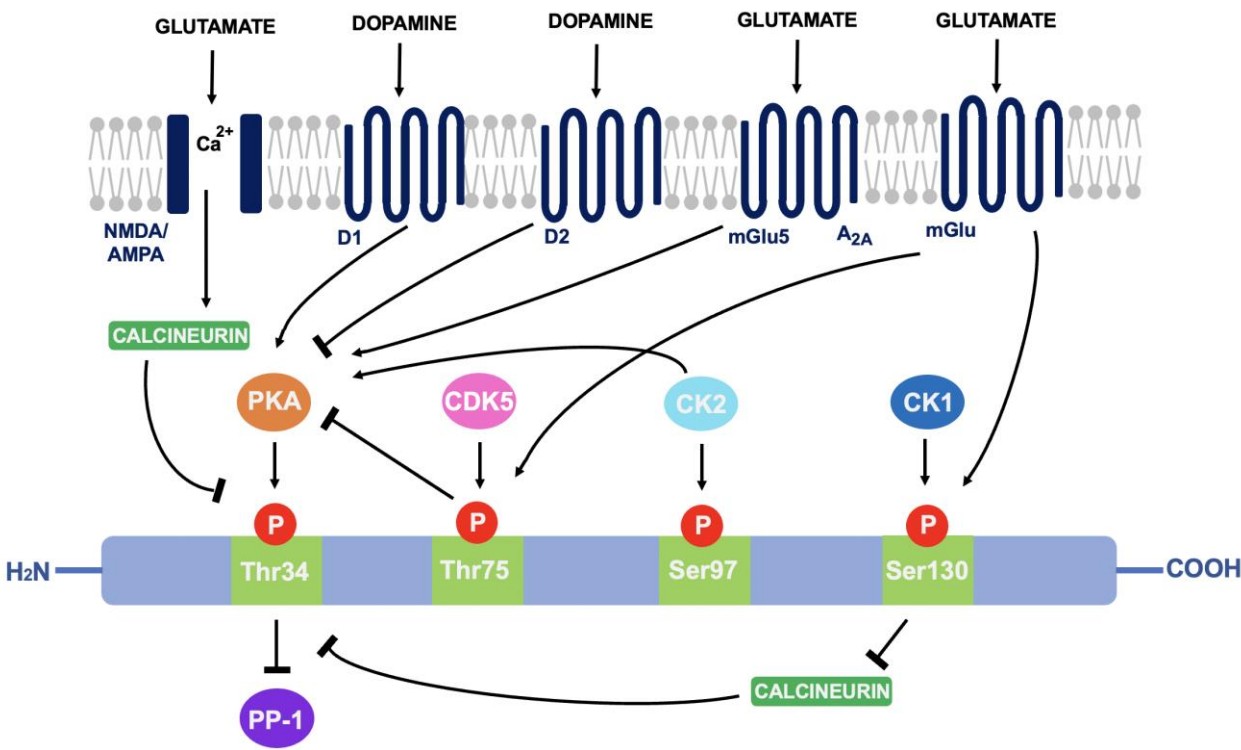

**Figure 3.** A summary of the actions of dopamine and glutamate on DARPP-32 phosphorylation: Through phosphorylation of DARPP-32 at four main amino acid sites ($Thr^{34}$, $Thr^{75}$, $Ser^{97}$ and $Ser^{130}$), DARPP-32 acts as either a PP-1 or PKA inhibitor.

Excitatory neurons from the cortex, thalamus and limbic areas of the brain input high levels of glutaminergic innervation to the (predominantly GABAergic) striatal neurons through both NMDA and non-NMDA classes of glutamate receptors [85]. The localisation of DARPP-32 within neurons expressing high levels of dopaminergic and glutaminergic innervation is indicative of the importance of these neurotransmitters in regulating DARPP-32. As both dopamine and glutamate neurotransmission is critical to addiction pathophysiology, it is also indicative of the involvement of DARPP-32 in substance abuse.

*4.4. DARPP-32 and Neuroplasticity*

Changes in neural plasticity discussed in Section 3 are strongly associated with a pathway known as the mitogen-activated protein kinase (MAPK), extracellular signal-related kinase (ERK) cascade [86]. This results in the activation of downstream transcription factors such as cAMP response element binding protein (CREB) and subsequent expression of proteins that cause structural dendritic and synaptic changes that are associated with all forms of substance abuse [87]. For example, opioids and psychostimulants have opposing effects on neuronal plasticity in response to this pathway. Opioids decrease the number and complexity of dendritic spines on MSNs and the prefrontal cortex, alongside hippocampus neurons and dopaminergic neurons in the VTA. Cocaine, methylphenidate and amphetamines do the reverse [88].

Dopaminergic and glutaminergic pathways modulate both DARPP-32 and the MAPK/ERK cascade, and ERK is a downstream effector of DARPP-32 [89]. Furthermore, PP-1 is usually responsible for activation of striatal-enriched tyrosine phosphatase (STEP); a phosphatase that dephosphorylates and deactivates ERK. Thus, through DARPP-32 dependent PP-1 inhibition, ERK is activated. ERK activation in response to d-amphetamine (an amphetamine salt), cocaine, morphine, THC and nicotine abuse was lacking in DARPP-

32 knock out (KO) mice, highlighting the importance of DARPP-32 involvement in this pathway [90].

### 4.5. DARPP-32 and Substances of Abuse

In addition to the integration of dopaminergic and glutaminergic transmission that makes DARPP-32 so relevant to addiction pathophysiology, the protein itself is also differentially influenced by the various drugs of abuse (see Figure 4).

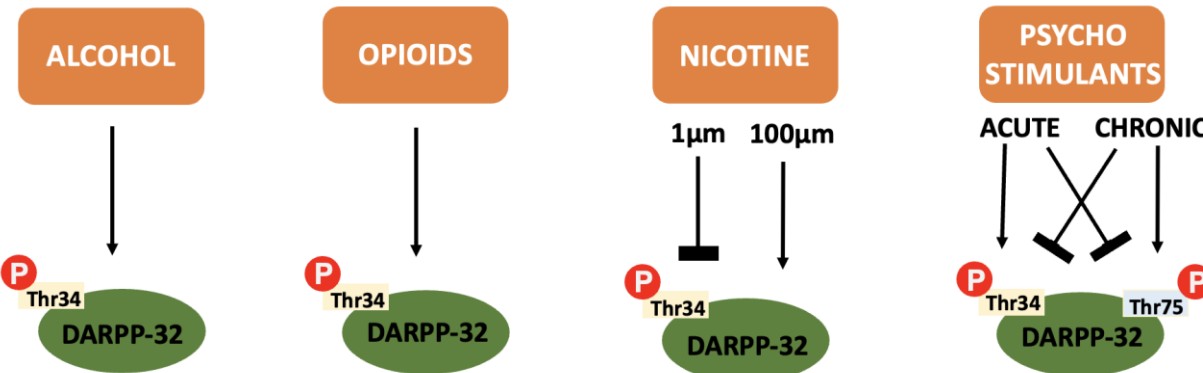

**Figure 4.** A summary of the effect of various drugs of abuse on DARPP-32 phosphorylation: Alcohol, opioids and cannabinoids cause an increase in $Thr^{34}$ independent of concentration or repeated administration. Nicotine causes inhibition of $Thr^{34}$ phosphorylation at low concentrations and increases $Thr^{34}$ phosphorylation at higher concentrations. Psychostimulants cause increased $Thr^{34}$ phosphorylation and decreased $Thr^{75}$ phosphorylation after acute administration and the reverse after chronic administration.

### 4.5.1. DARPP-32 and Alcohol

Moderate levels of alcohol (ethanol) are proven to increase $Thr^{34}$ phosphorylation, thus activating $D_1$ dependent cAMP/PKA/DARPP-32 signalling pathways and inhibiting PP-1 [91]. A downstream effect of this reduced phosphatase activity is phosphorylation of the $NR_1$ subunit on NMDA receptors [92]. Usually, ethanol is a potent inhibitor of NMDA receptors [93]. However, within DARPP-32 expressing brain regions via the mechanism discussed above, NMDA sensitivity to ethanol is reduced [94]. Disinhibition of NMDA receptors allows enhanced glutaminergic transmission that contributes to the reward pathway and allows long term synaptic plasticity to promote ethanol dependence. Somewhat expectantly, investigations into ethanol motivation using DARPP-32 KO mice proved it to be key in modulating ethanol-seeking behaviour [95].

Further studies into levels of DARPP-32 mRNA in rats with genetic preference or avoidance showed significant differences in genetic expression, implying the importance of DARPP-32 in genetic probability of addictive tendencies [96]. Another study showed that ethanol sensitized mice (who consumed more alcohol and were therefore more susceptible to addiction) had higher DARPP-32 phosphorylation when co-administered a $D_1$ receptor agonist [97]. This highlights a potential role of the $D_1$/DARPP-32/$Thr^{34}$ pathway in ethanol sensitization.

DARPP-32 therefore exists as a promising therapeutic target, due to prevalent involvement in ethanol dependence pathophysiology. Modulation of the phosphorylation pathways associated with it could reduce plasticity changes and ethanol reinforcement

### 4.5.2. DARPP-32 and Opioids

Some evidence shows that acute administration of opioids increases $D_1$ dependent phosphorylation of $Thr^{34}$ and has no effect on $Thr^{75}$ phosphorylation [98]. It is suggested that this $Thr^{34}$ phosphorylation augments hyperlocomotor responses to opioids, but seemingly has no effect on behavioural sensitisation [99].

Hyperlocomotion is a heightened state of locomotive activity; the forward progression carrying a person from one destination to the other [100]. It is often used as a phenotypical representation of substance abuse because all drugs of abuse have locomotor enhancing effects. Accordingly, increases in locomotor activity often parallel the progression of substance dependence, due to repeated administration progressively increasing the effect [101].

This dependence on DARPP-32 to cause hyperlocomotion in opioid use therefore supports the involvement of DARPP-32 in addiction progression. However, there are discrepancies in this knowledge. Locomotor activity is increased through opioid interaction with μ opioid receptors [102]. In striatonigral neurons within the striatum, activation of the μ receptor causes an interaction with $D_1$ receptors that inhibits the increase in DARPP-32 phosphorylation [103]. This would therefore reduce Thr$^{34}$ phosphorylation and contradict the effect of this on motor response. Supporting this, acute morphine administration to morphine-sensitised rats has shown a delayed increase in Thr$^{75}$ phosphorylation, hence PKA inhibition of and reduction in Thr$^{34}$ phosphorylation [104].

An interrelation of opioid effecting DARPP-32 and vice versa is clear. Further clarity of whether the mechanisms of this interaction progress addiction would clarify whether DARPP-32 is a potential target for opioid dependence treatment.

### 4.5.3. DARPP-32 and Nicotine

The effect of nicotine on DARPP-32 is dose-dependent, causing a sustained decrease in Thr$^{34}$ phosphorylation at low concentrations (1 μm) and transient increases at higher concentrations (100 μm). This is likely due to $D_2$ or $D_1$ receptor signalling at low or high concentrations, respectively [105]. In vivo arterial concentrations of nicotine are usually closer to the lower value; approximately 0.5 μm [106]. Hence, an educated guess would be to assume Thr$^{34}$ phosphorylation is low within human smokers.

Investigations using DARPP-32 KO mice displayed heightened nicotine intake, responsiveness to motor depressant effects and a generally enhanced behavioural response to nicotine [107]. It could therefore be hypothesised that low levels of Thr$^{34}$ phosphorylation could exert behavioural control of nicotine through its phosphorylation state. Modulation of DARPP-32 to influence this behavioural control would in this sense be a promising therapeutic target.

### 4.5.4. DARPP-32 and Cannabinoids

Evidence of the effect of cannabinoids on DARPP-32 phosphorylation are somewhat less concrete and explored than that of other substances of abuse. This is perhaps responsive to earlier discussions in Section 2.4 regarding the unacknowledged severity of CUD. Despite this, there is evidence to show that agonists for the $CB_1$ receptor (a neural cannabinoid receptor) do increase Thr$^{34}$ phosphorylation in MSNs [108].

This phosphorylation has been linked to the cataleptic effects of high dose cannabinoids [109]. Through genetic inactivation of receptors involved in the DARPP-32/PKA pathway and resulting decreases in motor depression, it is clear Thr$^{34}$ is involved in the suppressive psychomotor effects of cannabinoids [110]. To what extent this is relevant to DARPP-32 as a therapeutic target for addiction would require more understanding of DARPP-32 involvement in CUD pathophysiology.

Interaction between the $CB_1$ receptor and $D_2$ receptors to increase ERK phosphorylation and enhance $CB_1$ expression (thus increasing cannabinoid signalling) has been proven [111]. It is likely that DARPP-32 is an integrator within this cross talk and DARPP-32 shows promise as a potential target for CUD, but further clarifications are required.

### 4.5.5. DARPP-32 and Psychostimulants

Psychostimulants exhibit different degrees of DARPP-32 phosphorylation depending on acute or chronic administration. Acutely, Thr$^{34}$ phosphorylation is increased whilst Thr$^{75}$ phosphorylation decreases. After chronic administration, CDK-5 (and p53, another

transcription factor) are upregulated to result in reversal of this ratio [112]. DARPP-32 KO mice show reduced sensitivity, reward and locomotor activity acutely, and increased locomotor sensitivity after chronic use [113]. It is therefore plausible that these changes are dependent on the phosphorylation state of DARPP-32. This supports use as a therapeutic target.

Further investigations have highlighted the importance of DARPP-32 phosphorylation in psychostimulant dependence. Acutely, interrelations between cocaine, DARPP-32 and the ERK pathway have been realised. High levels of $Thr^{34}$ and decreased $Thr^{75}$ phosphorylation corresponds to increased ERK signalling, thus contributing to both genetic expression and behavioural response [114]. This enhanced signalling is associated with cocaine-conditioned place preference behaviour, which represents contextual drug reward [115]. Meanwhile, high levels of $Thr^{75}$ phosphorylation after chronic administration are intrinsically linked to psychostimulant-induced locomotion and behavioural sensitisation [116,117].

DARPP-32 is undoubtably integral to psychostimulant dependence. Arguments for application as a therapeutic target are most well-supported for psychostimulants as opposed to other abusable substance groups.

## 5. Discussion

Ultimately, it can be concluded that DARPP-32 is a modulator of the actions of abusable substances, and vice versa. When considering the joint reliance of addiction pathophysiology and DARPP-32 modulation on dopaminergic and glutaminergic signalling, this is unsurprising. Yet, little research has been carried out to use DARPP-32 in addiction treatment.

Genetic silencing of DARPP-32 via siRNA has yielded some promising results in the field of opiate addiction. SiRNA silencing was initially used to investigate the downstream effects on PP-1, ERK and CREB in response to DARPP-32 silencing [118]. This led to alterations in the activity of these effectors and facilitated further investigations using siRNA DARPP-32 to treat addiction. Intracerebral administration of gold nanorods complexed to DARPP-32 to decrease expression resulted in a lack of condition placed aversive behaviour in opiate-addicted animals in vivo [119,120]. This is an example of DARPP-32 influence with efficacious results in substance dependence reduction in an animal model. However, the technology has not been fully optimized and does not exist yet as an option for pharmacotherapy [121].

Methylphenidate has also been proven to induce changes in DARPP-32 expression; cerebral levels of DARPP-32 were altered in both young and adult rats after methylphenidate delivery [122]. Thus, reiterating the ability of DARPP-32 expression to be pharmacologically altered. In vivo pharmacological agents affecting other components in DARPP-32 pathways, for example $D_1$ agonists or antagonists, have demonstrated their ability to alter DARPP-32 phosphorylation [123]. Hence, it is entirely plausible that other pharmacological agents could influence this phosphorylation directly.

One aspect that must be carefully considered in lieu of this is the effect of phosphorylation targeting on functional levels of DARPP-32, particularly when considering previously discussed higher intakes of nicotine in KO mice. If targeting phosphorylation resulted in inactivation or degradation of DARPP-32, there could be an increased risk of nicotine uptake. This is relevant in the knowledge that nicotine is a commonly co-abused substance [124]. Compounds capable of selectively modulating phosphorylation without influencing total levels of DARPP-32 would therefore be of particular interest.

## 6. Conclusions

Current standards of treatment for substance abuse are universally incomplete, as evidenced by high mortality rates. Minimal and problematic pharmacological treatment options combined with the relentlessly persistent stigma associated with addiction are hugely detrimental. This is an important, global issue for which there is an evident unmet

clinical need. This review magnifies the relevance of DARPP-32 in addiction pathophysiology and potential application of DARPP-32 in the treatment of substance abuse. A compound capable of influencing DARPP-32 phosphorylation and resulting downstream effects (without negatively affecting functionality) could be therapeutically efficacious against substance abuse. With further investigation, a compound such as this could be implemented to improve upon substandard treatment options or address substance abuse disorders for which there are currently no licensed treatments.

**Author Contributions:** Writing—original draft preparation, M.R.G.; writing—review and editing, S.J.S.; All authors have read and agreed to the published version of the manuscript.

**Funding:** This work was supported by the funding body Wellcome Trust (grant code RS3684).

**Conflicts of Interest:** The authors declare no conflict of interest.

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
