# Peer review of "Exploring the Role of DARPP-32 in Addiction: A Review of the Current Limitations of Addiction Treatment Pathways and the Role of DARPP-32 to Improve Them"

_neurosci, doi:10.3390/neurosci3030035_

Round 1
Reviewer 1 Report
The review by Greener and Storr is very interesting and covers a topic of importance to the broad neuroscience audience.
A few minor comments are indicated that could improve the review manuscript.
In the section 2.1, While discussing the available medications, can the authors comment on how accessible these medications are to the individuals. Are they very expensive to buy etc.. (again supports the stigma issue discussed)
In the section 3, neurobiology of addiction is discussed. While brief, it may be important to direct the readers to reviews that discuss animal models of addiction, and in addition, briefly indicate in this section that drugs of abuse discussed in the previous section have overlapping and distinct neurobiological responses.
In section 5, it will be interesting for the authors to discuss the available options to silence or antagonize DARPP-32. There is some discussion on silencing, however it is not clear if selective pharmacotherapies are available, and if they have been tested in animal models of addiction.
Author Response
Hello,
Thank you very much for your insightful comments and useful feedback. I have made the following adjustments based on the suggestions:
Section 3: I have added an additional review on animal models for addiction and acknowledged that each substance will possess a different neurobiological response to this section. These statements are within the first and second paragraph of section 3.
Section 5: I have added statements that clarify whether pharmacotherapies are available and whether the technology discussed was tested in animal models.
Section 2.1: The suggestion to discuss treatment accessibility is a really interesting suggestion, and one that is relevant not only to AUD treatment but all substance abuse. However, as this is an international journal, it feels beyond the scope to discuss price/availability globally. For example, owing to the NHS in England treatment is readily available whereas in the US this would differ. Also, after investigating this as per your suggestion, lack of treatment for AUD disorder seems often to be a result of lack of perceived need from individuals or systemic issues in primary care/diagnosis. Again, this is a complex topic that requires lots of attention and discussion. I have therefore decided to keep sections 2.1-2.6 as issues with specific pharmacotherapies rather than broader issues with treatment accessibility or retention generally, owing to the pharmacological perspective of this review. However, in further studies and reviews this is a suggestion I will carefully consider, and I thoroughly appreciate the helpful suggestions.
Reviewer 2 Report
This review article by Greener and Storr highlighted an important healthcare issue with current standards of treatment for substance abuse. The review describes the available mechanistic approaches to therapeutically treat substance abuse and their limitations. The review also systematically explores the potential for DARPP-32 modulation as a therapeutic avenue under different substance abuse categories. Overall, the article is very well-written even for a lay person to comprehend, and can be published after addressing a few minor comments as follows:
- A short discussion on the relevance of targeting Thr34 phosphorylation of DARPP-32 and the levels of DARPP-32 might be added to the review based on the following:
Page 9, line 351: Moderate levels of alcohol are proven to increase Thr34 phosphorylation of DARPP-32. However, DARPP-32 KO mice displayed heightened nicotine intake (Page 10, line 399). Moreover, nicotine causes a sustained decrease in Thr34 phosphorylation at low concentrations (1µm) and transient increases at higher concentrations (100µm). In this scenario, careful consideration is to be given to modulating DARPP-32 phosphorylation status and the total levels of DARPP-32 protein itself for pharmacological interventions. For example, drugs that target Thr34 phosphorylation of DARPP-32 may affect the total functional levels of DARPP-32 (due to DARPP-32 inactivation/degradation) and thereby may rise the risk for nicotine uptake. This aspect is critical under conditions in which alcohol and nicotine abuse co-occur. - Based on the above perspective, the current statement in the conclusion “A compound capable of influencing DARPP-32 phosphorylation and resulting downstream effects could be applied as a therapeutic agent” can be refined. For instance, compounds capable of selectively modulating DARPP-32 Thr34 phosphorylation and its downstream effects with or without affecting other outcomes of DARPP-32 function/total levels DARPP-32 need to be screened for their therapeutic efficacy against substance abuses.
- There is a typo in the title of figure 1– “Glycogensis” =>> “Glycogenesis”
- Page 4, lines 186-194 has repetitive discussion. It should be re-written in concise words.
Author Response
Thank you for taking the time to review this article and for your extremely helpful feedback. Your first two points were particularly insightful and something I will heavily consider within future studies, thank you! I have incorporated the following changes to reflect the suggestions made:
- A discussion regarding phosphorylation targeting and total functional levels of DARPP-32 (especially in terms of co-abuse of nicotine as suggested) has been added to section 5
- The wording of this phrase in section 6 has been altered to consider the requirement for no loss of function.
- This typo has been corrected.
- Repetitive discussion at the end of section 2/start of section 3 has been corrected.
Best wishes
Megan Greener